# Single vs replicate Real-Time PCR SARS-CoV-2 testing: Lessons learned for effective pandemic management

**William R. Webb**[1], **Gauri Thapa**[1], **Alice Tirnoveanu**[2], **Sabrina Kallu**[3], **Charlene Loo Jin Yi**[3], **Nirali Shah**[3], **Joseph Macari**[3], **Sadie Mitchell**[3], **Graham J. Fagg**[3], **Rachael N. Jeremiah**[3], **Sandiya Theminimulle**[3], **Romina Vuono**[4‡]*, **Athina Mylona**[1,2‡]*

**1** Faculty of Medicine, Institute of Medical Sciences, Health and Social Care, Canterbury Christ Church University, Canterbury, United Kingdom, **2** Faculty of Natural and Applied Sciences, School of Psychology and Life Sciences, Canterbury Christ Church University, Canterbury, United Kingdom, **3** North Kent Pathology Service, Microbiology, Darent Valley Hospital, Dartford and Gravesham NHS Foundation Trust, Dartford, England, **4** Medway School of Pharmacy, University of Kent, Canterbury, England

‡ AM and RV are Joint Senior Authors
* athina.mylona@canterbury.ac.uk (AM); rv227@kent.ac.uk (RV)

**Data Availability Statement:** All relevant data are within the manuscript and its Supporting information files.

## Abstract

Coronavirus Disease 19 (COVID-19) caused by the SARS-CoV-2 virus remains a global pandemic having a serious impact on national economies and healthcare infrastructure. Accurate infection detection protocols are key to policy guidance and decision making. In this pilot study, we compared single versus replicate PCR testing for effective and accurate SARS-CoV-2 infection detection. One-Step Real-Time RT-PCR was employed for the detection of SARS-CoV-2 RNA isolated from individual nasopharyngeal swabs. A total of 10,014 swabs, sampled from the general public (hospital admissions, A&E, elective surgeries, cancer patients, care home residents and healthcare staff), were tested using standard replicate testing. Our analysis demonstrates that approximately 19% of SARS-CoV-2 infected individuals would have been reported as false negative if single sample Real-Time PCR testing was used. Therefore, two replicate tests can substantially decrease the risk of false negative reporting and reduce hospital and community infection rates. As the number of variants of concern increases, we believe that replicate testing is an essential consideration for effective SARS-CoV-2 infection detection and prevention of further outbreaks. A strategic approach limiting the number of missed infections is crucial in controlling the rise of new SARS-CoV-2 variants as well as the management of future pandemics.

## Introduction

COVID-19 caused by the Severe Acute Respiratory Syndrome Coronavirus-2 (SARS-CoV-2) remains a global pandemic with 174 million infected and 5,268,849 deaths (09/12/2021, source: WHO COVID-19 dashboard). SARS-CoV-2 was identified as the causal agent of unusual pneumonia cases in the city of Wuhan, P.R. China, in December 2019—January 2020 [1]. The

**Funding:** The author(s) received no specific funding for this work.

**Competing interests:** The authors have declared that no competing interests exist.

genetic sequence of SARS-CoV-2 was determined by Wu *et al.* on January 17[th] 2020 (NC_045512.2) [2]. Since then, at least 5,775 sequence variations have been reported and continue to rise [3]. Numerous potential coding targets for diagnostic use have been identified, including *Open Reading Frame 1 ab* (*ORF1ab*), *S* gene, *ORF3a* gene, *E* gene, *M* gene, *ORF6a* gene, *ORF7a* gene, *ORF7b*, *ORF8* gene, *N* gene and *ORF10* gene [4].

Currently, a range of genetic targets for diagnostic purposes have been utilised including the *ORF1ab*, *N* gene, *S* gene and *E* gene. The *N* gene sequence is conserved with other members of the corona virus family, therefore amplification of this region is utilised mainly as a guidance for re-testing with SARS virus sequence similarity of 90.25% (NC_004718.3) [5].

The specificity and accuracy of SARS-CoV-2 PCR diagnostic testing has been extensively considered in the last 19 months, as numerous perspectives in regard to the reporting of false negatives have been published [6–8]. Evidence of false negative reporting include cases of patients with COVID-19 specific symptoms and chest X-rays, a history of SARS-CoV-2 positive contacts but a negative nasopharyngeal swab PCR test result [9–12].

False negative reporting can significantly impact disease spread to close contacts and the community, especially for asymptomatic individuals [6]. Arevalo-Rodriguez *et al.* (2020) estimated false negative reporting between 2–29% [7]. This wide estimate is the product of a systematic review, which did not consider the sensitivity of PCR based SARS-CoV-2 detection protocols such as the number of replicates that should be carried out for nasopharyngeal swabs under investigation. In this pilot study, we have compared single versus replicate Real-Time PCR testing on a cohort of 10,014 nasopharyngeal swab samples.

## Materials and methods

### Nasopharyngeal swab collection

Nasopharyngeal swabs were tested at the North Kent Pathology Service (NKPS), Dartford and Gravesham NHS Trust. These were received form the Dartford and Gravesham NHS Trust, the Medway NHS Foundation Trust and the Oxleas NHS Foundation Trust (hospital admissions, A&E, elective surgeries, cancer patients, staff), as well as Dartford and Medway care home residents. Upon sampling each nasopharyngeal swab was placed in a sterile transport media tube and was kept chilled during transport and prior to viral RNA extraction. The nasopharyngeal swabs were processed within 24 hours from collection.

### SARS-CoV-2 RNA extraction

Total RNA was extracted using HigherPurity™ Viral RNA extraction kit (Cat. No: AN0805-XL) from Canvax Biotech in accordance with the manufacturers protocol. Total RNA was eluted into 40μl of RNAse free water. Total RNA (duplicate testing) was tested immediately after extraction.

### One-Step Real-Time RT-PCR SARS-CoV-2 detection

Two commercial quantitative One-Step Real-Time RT-PCR SARS-CoV-2 detection kits were used: (1) Vitassay SARS-CoV-2, Real-time PCR detection kit, Vitassay Healthcare, S.L.U. Huesca, Spain, (2) Primerdesign™ LTD, Coronavirus (COVID-19) Genesig® Real-Time PCR assay, Primerdesign Ltd, Chandlers Ford, Southampton, U.K. Both kits detect SARS-CoV-2 based on the SARS-CoV-2 Wuhan genome (NCBI Taxonomy ID:269049, REFSeq: NC_045512.2). One-Step Real-Time RT-PCR reactions were prepared in accordance with the manufacturers' protocols. Internal controls were included in all RT-PCR reactions, in accordance with the manufacturers' protocols in order to confirm the accuracy of each RT-PCR

reaction. All quantitative PCR reactions (single and duplicate testing) were carried out on the Biorad CFX96 Real-Time PCR detection system and for both assay kits the total number of cycles was 45 (according to the manufacturers' protocols).

The RT-PCR data is available as a supplementary document.

### Ethical approval

Ethical approval is not applicable for this study. Patients were not specifically recruited for this study. This is a study of testing methodology only and the outcome of SARS-CoV-2 RT-PCR testing was analysed with no link to patient clinical information.

## Results

In order to evaluate the effectiveness of a SARS-CoV-2 testing protocol that minimizes the frequency of false negative test results we employed a duplicate PCR reaction approach per tested nasopharyngeal swab. All RT-PCR reactions that resulted in a Ct value lower than 40 cycles and a clear amplification curve, were recorded as positive.

A total of 10,014 nasopharyngeal swabs were tested using 20,028 duplicate Real-Time PCR reactions. The tested nasopharyngeal swabs were collected between the months of May and July 2020 and the collection approach was not restricted to COVID-19 symptomatic or SARS-CoV-2 infected asymptomatic individuals. Therefore, our nasopharyngeal swab collection method accurately represents the public infection rates at the time of sampling. Primerdesign One-Step Real-Time RT-PCR assay was used for 18,672 tests and the Vitassay One-Step Real-Time RT-PCR assay was used for 1,356 tests. Out of 10,014 patients tested, 491 (2.45%) were reported positive with both assay kits (Fig 1A). Further analysis of the positive results showed that 302 or 61.5% of these were reported as duplicate positives (positive SARS-CoV-2 RNA detection on both replicate 1 and 2), 85 or 17.3% of the positive results were only detected on replicate 1 and 104 or 21.2% of the positive reported samples were only detected on replicate 2 (Fig 1A).

Out of the 10,014 patients, 678 (corresponding to 1,356 duplicate Real-Time RT-PCR reactions) were tested with the Vitassay One-Step Real-Time RT-PCR kit. Of these, 27 patients (3.98%) were reported as positive, with 23 (85% of all positive patients) detected as duplicate positives (detection was observed in both replicates). In addition, 1 patient was detected as positive on replicate 1 only and 3 patients were detected as positive on replicate 2 only. Therefore, 4 patients were detected positive on one replicate (15% of all positive patients).

The Primerdesign One-Step Real-Time RT-PCR assay was used to test 9,336 patients (corresponding to 18,672 duplicate Real-Time RT-PCR reactions). Of these, a total of 464 patients (4.97%) were detected as SARS-CoV-2 positive, with 279 patients (60% of all positive patients) detected as duplicate positive (detection on replicate 1 and replicate 2), 84 patients detected positive on replicate 1 only and 101 patients detected as positive on replicate 2 only. Overall, 185 patients were detected positive on one replicate (40% of all positive patients).

Fig 1B–1D summarizes the reporting profile for Real-Time PCR SARS-CoV-2 testing. Testing of 10,014 nasopharyngeal patient swabs has shown that 61.5% of SARS-CoV-2 positive patients were detected on double replicate positive PCR reactions, while 17.3% were detected positive on replicate 1 and 21.2% were detected positive on replicate 2. Overall, we can conclude that approximately 19% of patients were detected as SARS-CoV-2 positive by only one of the two PCR reaction replicates and these would be missed if single reaction testing was employed.

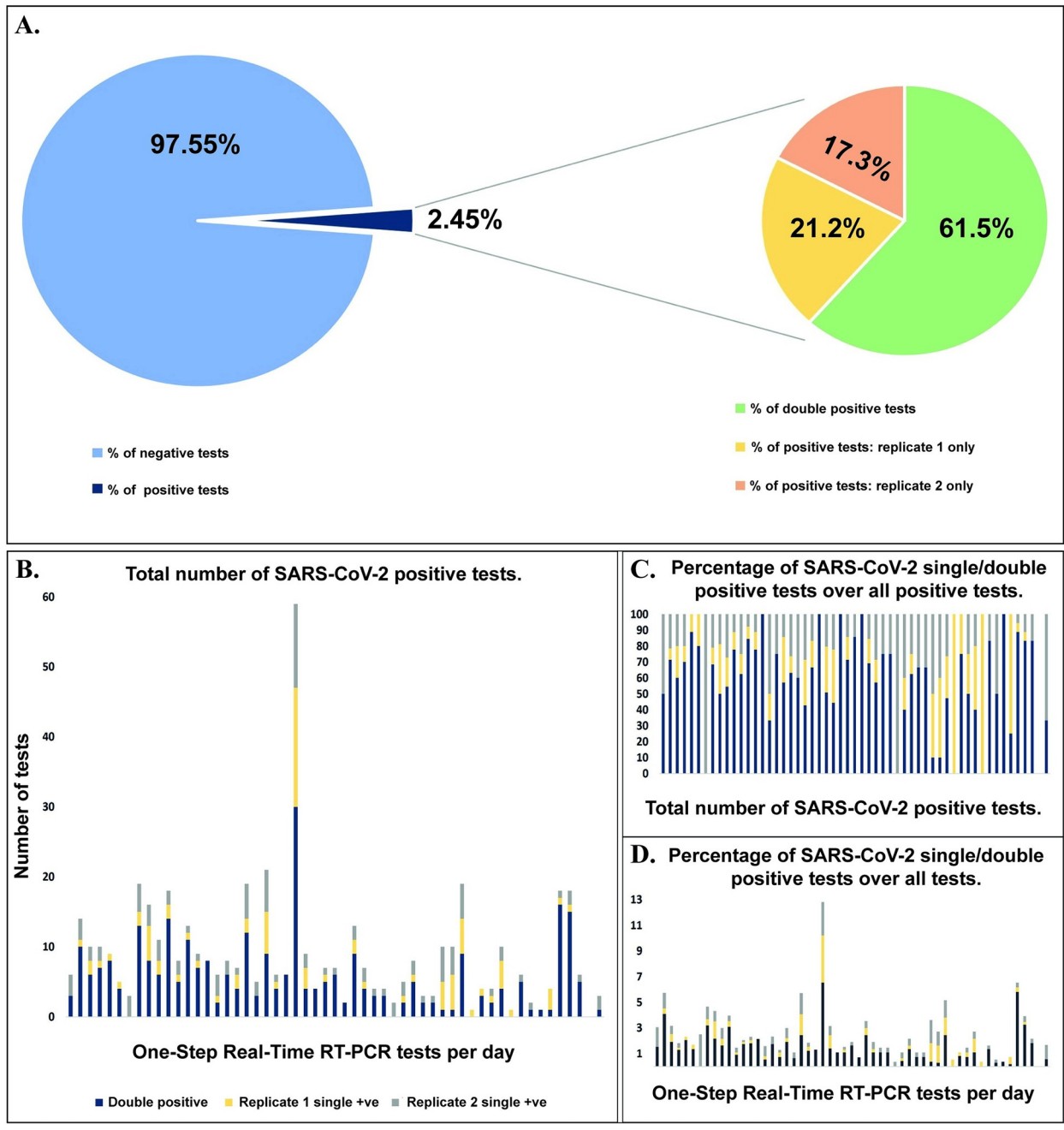

**Fig 1. Graphic representation of the SARS-CoV-2 testing outcome of 10,014 nasopharyngeal swabs. (A)** Graphic representation of SARS-CoV-2 positive sample reporting relative to Real-Time PCR reaction outcome. 491 patients or 2.45% tested positive for the SARS-CoV-2 virus. While 61.5% of SARS-CoV-2 infections were detected in both Real-Time PCR repeat testing reactions almost 40% of SARS-CoV-2 infections were detected in only one of the two repeat Real-Time PCR testing reactions. **(B)** Total number of patient nasopharyngeal swab tests detected positive for SARS-CoV-2 on duplicate, replicate 1 or replicate 2 Real-Time PCR. **(C)** Graphic representation of the percentage of duplicate, replicate 1 only and replicate 2 only Real-Time PCR confirmed SARS-CoV-2 positive samples over all positive detected samples. **(D)** Graphic representation of the percentage of duplicate, replicate 1 only and replicate 2 only Real-Time PCR confirmed SARS-CoV-2 positive samples over all tested samples.

## Discussion

In this study we examined whether replicate RT-PCR testing of SARS-CoV-2 RNA, isolated from nasopharyngeal swabs, increases the reliability of testing results' reporting. We tested

10,014 nasopharyngeal swabs, which equates to 20,028 duplicate Real-Time PCR reactions and identified 491 SARS-CoV-2 positive patients. As mentioned above, RT-PCR reactions were recorded as positive when a Ct value lower than 40 cycles was detected and a clear amplification curve was confirmed. This applies to single and duplicate RT-PCR reactions.

Our findings demonstrate that a singlet testing policy would have failed to detect approximately 19% of SARS-CoV-2 positive patients, approximately 94 in 10,000 patients and a predictive ratio of 940/100,000 individuals. This level of false negatives, in correlation with an $R_0$ number >1, can contribute to sustained and increasing SARS-CoV-2 infection rates within hospitals and the wider community.

False negative PCR outcomes can be influenced by substandard sampling and storage leading to inefficient viral RNA isolation and detection. In addition, a false negative PCR can be due to low viral copy number which can either reflect the end of the infection or, more worryingly, the early infection stage. We did not investigate these factors as they fell outside the scope of the present study, which instead focused on single vs duplicate testing and the rate of false negative outcomes. Our conclusion is in agreement with a systematic review conducted by Arevalo-Rodriguez *et al.* (2020) estimating a false negative reporting ranging between 2–29% [7] and shows that regardless of the cause for a false negative PCR, duplicate testing increases the accuracy of SARS-CoV-2 infection reporting and can minimise the chance of infection outbreaks. Moreover, our work complements the analysis carried out by Arevalo-Rodriguez *et al.* as to our knowledge, this is the first study that determines the proportion of PCR tested nasopharyngeal swabs that show repeatable results in replicate testing and considers whether duplicate positive or individual replicate positive testing can reliably detect the presence of SARS-CoV-2 viral RNA and consequently infection.

We acknowledge that duplicate testing may be prohibited due to its cost and practical considerations especially when the daily infection rate is high, but with several variants of concern arising, it is imperative to have a strategic methodology that limits the number of missed infections. Therefore, a duplicate testing protocol should be employed for accurate pandemic modelling management and prevention of infection outbrakes. This is crucial not only for this pandemic, which continues to change and develop but future pandemics as well.

## Supporting information

**S1 Table. One-Step Real-Time RT-PCR SARS-CoV-2 detection raw data.** The table summarizes the outcome of each of the 20,028 One-Step Real-Time RT-PCR reaction tests for the detection of SARS-CoV-2, carried out between May and July 2020.
(XLSX)

## Acknowledgments

The authors would like to thank Mrs Tina Bailey and the North Kent Pathology Service, Darent Valley Hospital, Dr. Susan Plummer (Institute of Medical Sciences, Canterbury Christ Church University), Professor Susan Barker and Dr. Gurprit S. Lall (Medway School of Pharmacy, Universities of Kent and Greenwich) for their unconditional support.

## Author Contributions

**Conceptualization:** William R. Webb, Romina Vuono, Athina Mylona.

**Formal analysis:** William R. Webb, Romina Vuono, Athina Mylona.

**Investigation:** William R. Webb, Gauri Thapa, Alice Tirnoveanu, Sabrina Kallu, Charlene Loo Jin Yi, Nirali Shah, Joseph Macari, Sadie Mitchell, Graham J. Fagg, Romina Vuono, Athina Mylona.

**Project administration:** Romina Vuono.

**Resources:** Rachael N. Jeremiah, Sandiya Theminimulle.

**Supervision:** Romina Vuono, Athina Mylona.

**Writing – original draft:** William R. Webb, Athina Mylona.

**Writing – review & editing:** Romina Vuono, Athina Mylona.

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
