## [Decision Letter · Decision Letter 0]

16 Mar 2022

PONE-D-21-40437Single vs replicate Real-Time PCR SARS-CoV-2 testing: lessons learned for effective pandemic management.PLOS ONE

Dear Dr. Mylona,

Thank you for submitting your manuscript to PLOS ONE. After careful consideration, we feel that it has merit but does not fully meet PLOS ONE’s publication criteria as it currently stands. Therefore, we invite you to submit a revised version of the manuscript that addresses the points raised during the review process.

Authors need to prepare responses to reviewers' comments and the text of the manuscript in accordance with these comments.

We look forward to receiving your revised manuscript.

Kind regards,

Ruslan Kalendar

Academic Editor

PLOS ONE

Reviewers' comments:

Reviewer's Responses to Questions

**Comments to the Author**

1. Is the manuscript technically sound, and do the data support the conclusions?

Reviewer #1: Yes

Reviewer #2: Yes

Reviewer #3: Partly

2. Has the statistical analysis been performed appropriately and rigorously? 

Reviewer #1: Yes

Reviewer #2: Yes

Reviewer #3: No

3. Have the authors made all data underlying the findings in their manuscript fully available?

Reviewer #1: Yes

Reviewer #2: Yes

Reviewer #3: No

4. Is the manuscript presented in an intelligible fashion and written in standard English?

Reviewer #1: Yes

Reviewer #2: Yes

Reviewer #3: Yes

5. Review Comments to the Author

Reviewer #1: 

In this report, the authors compare the accuracy estimates of two SARS-CoV-2 testing protocols, Vitassay One-Step Real-Time RT-PCR and Primerdesign One-Step Real-Time RT-PCR assay, specifically showing the need for replicate testing considering their accuracies. While the paper presents results that are scientifically sound and reiterate previously published accuracy estimates of SARS-CoV-2 tests, it does not address a sufficiently original research question to merit publication in PLoS One. Most diagnostic tests have non-zero error rates which means accuracy will be better with replicate testing. The paper doesn’t add novel value as to situations in the ongoing COVID-19 pandemic where replicate testing will be specifically necessary, eg. certain variants, populations, or viral loads.

Reviewer #2: 

Interesting confirmatory paper in real field on the high risk of false negative results for SARS-CoV-2 RNA detection in clinical samples. Two different PCR assays were used on high number of clinical samples. The paper is short and well written.

Minor points

P5. Give the periods of inclusion.

P5. Please indicate the percentages in paragraphs 3 and 4.

P8. Update the reference 7:

Arevalo-Rodriguez I, Buitrago-Garcia D, Simancas-Racines D, Zambrano-Achig P, Del Campo R, Ciapponi A, Sued O, Martinez-García L, Rutjes AW, Low N, Bossuyt PM, Perez-Molina JA, Zamora J. False-negative results of initial RT-PCR assays for COVID-19: A systematic review. PLoS One. 2020 Dec 10;15(12):e0242958.

Reviewer #3: 

Summary:

The authors us data from real-time PCR testing of 10,014 nasopharangeal swab specimens to compare the rate of detection of SARS-CoV-2 in single vs replicate testing scenarios. The authors make the argument that replicate testing provides a superior detection mechanism for ensuring accurate test results for pandemic control. Of the original 10,014 swabs, 9336 were tested twice using the Vitassay One-Step Real-Time RT-PCR assay, which uses 2 SARS-CoV-2 specific primer/probe set; and 678 swabs were tested using the Primerdesign LTD, Coronavirus (COVID-19) Genesig Real-Time PCR assay, which uses 1 SARS-CoV-2 specific primer/probe set. The authors report that, in aggregate, 2.45% of samples tested positive on either the first, second, or both replicates. Of these, 17.3% tested positive on only the second replicate, indicating that a non-replicate testing method would have missed a substantial fraction of positive cases.

Major concerns:

• More information is needed on sample selection within the Materials and Methods. Were these swabs collected from symptomatic, suspected positive cases, routine screening of individuals seeking medical care for all maladies, asymptomatic surveillance, or some other reason? This is important for understanding the relevance of the work to pandemic management.

• It is not clear from the methods section and the Ethics Statement that this work has undergone appropriate institutional ethics review. The manner in which patient samples were obtained, the sample identities anonymised, and sample identities tracked and recorded through the replicate testing process was not indicated. If this is a chart review, this should be stated and the appropriate research ethics approvals or exemptions cited. The authors should provide confirmation of ethical review or exemption in accordance with journal and institutional standards: "Any research involving personal information, whether identifiable or anonymised, must be approved via full or proportionate review by an appropriate Research Ethics Committee." Section 7.6.1: https://www.canterbury.ac.uk/asset-library/research/Governance-and-Ethics/code-of-conduct.pdf

• Two different testing platforms were used, with different rates of positivity and criteria for assigning a sample as ‘positive,’ yet the data were aggregated. Authors should explain why data aggregation was appropriate in this case, or they should separate the data according to testing platform.

• A major weakness in the interpretation of the results stems from the binary nature of the report of ‘positive’ or ‘negative.’ Depending on how a Cq value cutoff was applied, replicate tests falling near the Cq threshold may be separately categorized as positive or negative, despite experimental results that are almost identical. The authors should examine and/or discuss the degree to which replicate agreement is affected by the application of a Cq cutoff.

• Authors have indicated that data are freely available but have not indicated the location or means of obtaining the data.

Minor concerns:

• The duration and method of RNA storage between swab acquisition and testing, and between testing replicates should be described.

• It appears from the methods that replicate tests were performed on the same platform as the initial test, but this should be explicitly stated.

• In the end of the results section, the authors write, “Testing of 10,014 nasopharyngeal patient swabs has shown that 61.5% of SARS-CoV-2 positive patients were detected on double replicate positive PCR reactions, while approximately 19% of patients were detected as SARS-CoV-2 positive by only one out of the two PCR reaction replicates.” This statement appears to be made in error, since the two values do not add up to 100%. Authors should revisit the intent of this sentence and edit accordingly.

• The word ‘outbreaks’ is misspelled in the abstract.

• “corona virus” should be a single word in the Introduction.

6. PLOS authors have the option to publish the peer review history of their article (what does this mean?). If published, this will include your full peer review and any attached files.

Reviewer #1: No

Reviewer #2: **Yes: **Laurent Belec

Reviewer #3: No

---

## [Author Response · Author response to Decision Letter 0]

2 May 2022

Dear Editor,

we thank you for the review process and the reviewers' comments. 

A detailed "Response to reviewers" letter has been prepared and submitted for revision together with the revised manuscript with and without track changes. 

We will be looking forward to your response. 

Kind regards

Athina Mylona

---

## [Decision Letter · Decision Letter 1]

30 May 2022

Single vs replicate Real-Time PCR SARS-CoV-2 testing: lessons learned for effective pandemic management.

PONE-D-21-40437R1

Dear Dr. Mylona,

We’re pleased to inform you that your manuscript has been judged scientifically suitable for publication and will be formally accepted for publication once it meets all outstanding technical requirements.

Kind regards,

Ruslan Kalendar

Academic Editor

PLOS ONE

---

## [Editor Report · Acceptance letter]

6 Jul 2022

PONE-D-21-40437R1 

Single vs replicate Real-Time PCR SARS-CoV-2 testing: lessons learned for effective pandemic management.  

Dear Dr. Mylona:

I'm pleased to inform you that your manuscript has been deemed suitable for publication in PLOS ONE. Congratulations! Your manuscript is now with our production department. 

Kind regards, 

on behalf of

Professor Ruslan Kalendar 

Academic Editor

PLOS ONE